# END-TO-END LEARNING OF ENERGY-BASED REPRESENTATIONS FOR IRREGULARLY-SAMPLED SIGNALS AND IMAGES

## ABSTRACT

For numerous domains, including for instance earth observation, medical imaging, astrophysics,..., available image and signal datasets often irregular space-time sampling patterns and large missing data rates. These sampling properties is a critical issue to apply state-of-the-art learning-based (e.g., auto-encoders, CNNs,...) to fully benefit from the available large-scale observations and reach breakthroughs in the reconstruction and identification of processes of interest. In this paper, we address the end-to-end learning of representations of signals, images and image sequences from irregularly-sampled data, *i.e.* when the training data involved missing data. From an analogy to Bayesian formulation, we consider energy-based representations. Two energy forms are investigated: one derived from auto-encoders and one relating to Gibbs energies. The learning stage of these energy-based representations (or priors) involve a joint interpolation issue, which resorts to solving an energy minimization problem under observation constraints. Using a neural-network-based implementation of the considered energy forms, we can state an end-to-end learning scheme from irregularly-sampled data. We demonstrate the relevance of the proposed representations for different case-studies: namely, multivariate time series, 2 images and image sequences.

## 1 INTRODUCTION

In numerous application domains, the available observation datasets do not involve gap-free and regularly-gridded signals or images. The irregular-sampling may result both from the characteristics of the sensors and sampling strategy, e.g. considered orbits and swaths in spacebone earth observation and astrophysics, sampling schemes in medical imaging, as well as environmental conditions which may affect the sensor, e.g. atmospheric conditions and clouds for earth observation.

A rich literature exists on interpolation for irregularly-sampled signals and images (also referred to as inpainting in image processing (4)). A classic framework states the interpolation issue as the miminisation of an energy, which may be interpreted in a Bayesian framework. A variety of energy forms, including Markovian priors (12), patch-based priors (20), gradient norms in variational and/or PDE-based formulations (4), Gaussian priors () as well as dynamical priors in fluid dynamics (3). The later relates to optimal interpolation and kriging (8), which is among the state-of-the-art and operational schemes in geoscience (10). Optimal schemes classically involve the inference of the considered covariance-based priors from irregularly-sampled data. This may however be at the expense of Gaussianity and linearity assumptions, which do not often apply for real signals and images. For the other types of energy forms, their parameterization are generally set a priori and not learnt from the data. Regarding more particularly data-driven and learning-based approaches, most previous works (2; 11; 20) have addressed the learning of interpolation schemes under the assumption that a representative gap-free dataset is available. This gap-free dataset may be the image itself (9; 20; 18). For numerous application domains, as mentionned above, this assumption cannot be fulfilled. Regarding recent advances in learning-based schemes, a variety of deep learning models, e.g. (7; 16; 24; 23), have been proposed. Most of these works focus on learning an interpolator. One may however expect to learn not only an interpolator but also some representation of considered data, which may be of interest for other applications. In this respect, RBM models (Restricted Boltzmann

Machines) (22; 6) are particularly appealing at the expense however of computationally-expensive MCMC schemes.

In this paper, we aim to learn representations of signals or images from irregularly-sampled observation datasets. Our contribution is three-fold:

- an end-to-end learning of energy-based representations from irregularly-sampled training data. Based on a neural-network architecture, it jointly embeds the considered energy form and an associated interpolation scheme.
- besides classic auto-encoder representations, we introduce NN-based Gibbs-Energy representations, which relate to Markovian priors embedded in CNNs.
- the demonstration of the relevance of the proposed end-to-end learning framework for different data types, namely time series, images and image sequences, with possibly very high missing data rates.

The remainder is organized as follows. Section 2 formally states the considered issue. We introduce the proposed end-to-end learning scheme in Section 3. We report numerical experiments in Section 4 and discuss our contribution with respect to related work in Section **??**.

## 2 PROBLEM STATEMENT

In this section, we formally introduce the considered issue, namely the end-to-end learning of representations and interpolators from irregularly-sampled data. Within a classic Bayesian or energy-based framework, interpolation issues may be stated as a minimization issue

$$\widehat{X} = \arg\min_X U_\theta(X) \text{ subject to } X_\Omega = Y_\Omega \tag{1}$$

where $X$ is the considered signal, image or image series (referred to hereafter as the hidden state), $Y$ the observation data, only available on a subdomain $\Omega$ of the entire domain $\mathcal{D}$, and $U_\theta()$ the considered energy prior parameterized by $\theta$. As briefly introduced above, a variety of energy priors have been proposed in the literature, e.g. (4; 20; 5).

We assume we are provided with a series of irregularly-sampled observations, that is to say a set $\{Y^{(i)}, \Omega^{(i)}\}_{i\in\{1,...,N\}}$, such that $\Omega^{(i)} \subset \mathcal{D}$ and $Y^{(i)}$ is only defined on subdomain $\Omega^{(i)}$. Assuming that all $X^{(i)}$ share some underlying energy representation $U_\theta()$, we may define the following operator $\mathcal{I}$

$$\mathcal{I}\left(U_\theta, Y^{(i)}, \Omega^{(i)}\right) = \arg\min_X U_\theta(X) \text{ subject to } X_{\Omega^{(i)}} = Y^{(i)}_{\Omega^{(i)}} \tag{2}$$

such that $\mathcal{I}(Y^{(i)}, \Omega^{(i)}) = X^{(i)}$. Here, we aim to learn the parameters $\theta()$ of the energy $U_\theta()$ from the available observation dataset $\{Y^{(i)}, \Omega^{(i)}\}_i$. Assuming operator $\mathcal{I}$ is known, this learning issue can be stated as the minimization of reconstruction error for the observed data

$$\widehat{\theta} = \arg\min_\theta \sum_i \left\| Y^{(i)} - \mathcal{I}\left(U_\theta, Y^{(i)}, \Omega^{(i)}\right) \right\|_{\Omega^{(i)}}^2 \tag{3}$$

where $\|.\|_\Omega^2$ refers to the L2 norm evaluated on subdomain. Learning energy $U_\theta()$ from observation dataset $\{Y^{(i)}, \Omega^{(i)}\}_i$ clearly involves a joint interpolation issue solved by operator $\mathcal{I}$.

Given this general formulation, the end-to-end learning issue comes to solve minimization (3) according to some given parameterization of energy $U_\theta()$. In (3), interpolation operator $\mathcal{I}$ is clearly critical. In Section 3, we investigate a neural-network implementation of this general framework, which embeds a neural-network formulations both for energy $U_\theta()$ and interpolation operator $\mathcal{I}$.

## 3 PROPOSED END-TO-END LEARNING FRAMEWORK

In this section, we detail the proposed neural-network-based implementation of the end-to-end formulation introduced in the previous section. We first present the considered parameterizations for energy $U_\theta()$ in (3) (Section 3.1). We derive associated NN-based interpolation operators $\mathcal{I}$ (Section 3.2) and describe our overall NN architectures for the end-to-end learning of representations and interpolators from irregularly-sampled datasets (Section 3.3).

### 3.1 NN-BASED ENERGY FORMULATION

We first investigate NN-based energy representations based on auto-encoders (15). Let us denote by $\phi_E$ and $\phi_D$ the encoding and decoding operators of an auto-encoder (AE), which may comprise both dense auto-encoders (AEs), convolutional AEs as well as recurrent AEs when dealing with time-related processes. The key feature of AEs is that the encoding operator $\phi_E$ maps the state $X$ into a low-dimensional space. Auto-encoders are naturally associated with the following energy

$$U_\theta(X) = \|X - \phi_D(\phi_E(X))\|^2 \tag{4}$$

Minimizing (1) according to this energy amounts to retrieving the hidden state whose low-dimensional representation in the encoding space matches the observed data in the original decoded space. Here, parameters $\theta$ refer to the parameters of the encoder $\phi_E$ and decoder $\phi_D$, respectively $\theta_E$ and $\theta_D$.

The mapping to lower-dimensional space may be regarded as a potential loss in the representation potential of the representation. Gibbs models provide an appealing framework for an alternative energy-based representation, with no such dimensionality reduction constraint. Gibbs models introduced in statistical physics have also been widely explored in computer vision and pattern recognition (13) from the 80s. Gibbs models relate to the decomposition of $U_\theta$ as a sum of potentials $U_\theta(X) = \sum_{c \in \mathcal{C}} V_c(X_c)$ where $\mathcal{C}$ is a set of cliques, *i.e.* a set of interacting sites (typically, local neighbors), and $V_c$ the potential on clique $c$. In statistical physics, this formulation states the global energy of the system as the sum of local energies (the potential over locally-interacting sites). Here, we focus on the following parameterization of the potential function

$$U_\theta(X) = \sum_{s \in \mathcal{D}} \|X_s - \psi(X_{\mathcal{N}_s})\|^2 \tag{5}$$

with $\mathcal{N}_s$ the set of neighbors of site $s$ for the entire domain $\mathcal{D}$ and $\psi$ a potential function. Low-energy state for this energy refers to state $X$ which operator $\psi$ provides a good prediction at any site $s$ knowing the state in the neighborhood $\mathcal{N}_s$ of $s$. This type of Gibbs energy relates to Gaussian Markov random fields, where the conditional likelihood at one site given its neighborhood follows a Gaussian distribution. We implement this type of Gibbs energy using the following NN-based parameterization of operator $\psi$:

$$\psi(X) = \psi_2(\psi_1(X)) \tag{6}$$

It involves the composition of a space and/or time convolutional operator $\psi_1$ and a coordinate-wise operator $\psi_2$. The convolutional kernel for operator $\psi_1$ is such that the coefficients for the center of convolutional window are set to zero. This property fulfills the constraint that $X(s)$ is not involved in the computation of $\psi(X_{\mathcal{N}_s})$ at site $s$. As an example, for a univariate image, $\psi_1$ can be set as a convolutional layer with $N_F$ filters with kernels of size 3x3x1, such that for each kernel $K_f$ $K_f(1,1,0) = 0$ (the same applies to biases). In such a case, operator $\psi_2$ would be a convolution layer with one filter with a kernel of size 1x1x$N_F$. Both $\psi_1$ and $\psi_2$ can also involve non-linear activations. Without loss of generality, given this parameterization for operator $\psi$, we may rewrite energy $U_\theta$ as $U_\theta(X) = \|X - \psi(X)\|^2$ where $\psi(X)$ at site $s$ is given by $\psi(X_{\mathcal{N}_s})$.

Overall, we may use the following common formulation for the two types of energy-based representation

$$U_\theta(X) = \|X - \psi(X)\|^2 \tag{7}$$

They differ in the parameterization chosen for operator $\psi$.

### 3.2 NN-BASED INTERPOLATOR

Besides the NN-based energy formulation, the general formulation stated in (3) involves the definition of interpolation operator $\mathcal{I}$, which refers to minimization (1). We here derive NN-based interpolation architectures from the considered NN-based energy parameterization.

Given parameterization (7), a simple fixed-point algorithm may be considered to solve for (3). This algorithm at the basis of DINEOF algorithm and XXX for matrix completion under subspace constraints (2; 14) involves the following iterative update

$$\begin{aligned}
X_p^{(k+1)} &= \psi(X^{(k)}) \\
X^{(k+1)}(\Omega) &= Y(\Omega) \\
X^{(k+1)}(\overline{\Omega}) &= X_p^{(k+1)}(\overline{\Omega})
\end{aligned} \tag{8}$$

Interestingly, the algorithm is parameter-free and can be readily implemented in a NN architecture given the number of iterations to be considered.

Given some initialisation, one may typically consider an iterative gradient-based descent which applies at each iteration $k$

$$
\begin{array}{rcl}
X_p^{(k+1)} & = & X^{(k)} - \lambda J_{U_\theta}\left(X^{(k)}\right) \\
X^{(k+1)}\left(\Omega\right) & = & Y\left(\Omega\right) \\
X^{(k+1)}\left(\overline{\Omega}\right) & = & X_p^{(k+1)}\left(\overline{\Omega}\right)
\end{array}
\tag{9}
$$

with $J_{U_\theta}$ the gradient of energy $U_\theta$ w.r.t. state $X$, $\lambda$ the gradient step and $\overline{\Omega}$ the missing data area. Automatic differentiation tool embedded in neural network frameworks may provide the numerical computation for gradient $J_{U_\theta}$ given the NN-based parameterization for energy $U_\theta$. This proved numerically too expensive and was not further investigated in our experiments. Given the considered form for energy $U_\theta$, its gradient w.r.t. $X$ decomposes as a product

$$
J_{U_\theta}\left(X\right) = J_\psi\left(X\right)\left(X - \psi\left(X\right)\right)
\tag{10}
$$

and $X - \psi\left(X\right)$ may be regarded as a suboptimal gradient descent. Hence, rather than considering the true Jacobian $J_\psi$ for operator $\psi$, we may consider an approximation through a trainable CNN $G()$ such that the gradient descent becomes

$$
\begin{array}{rcl}
X_p^{(k+1)} & = & X^{(k)} - G\left(X^{(k)}, \psi\left(X^{(k)}\right)\right) \\
X^{(k+1)}\left(\Omega\right) & = & Y\left(\Omega\right) \\
X^{(k+1)}\left(\overline{\Omega}\right) & = & X_p^{(k+1)}\left(\overline{\Omega}\right)
\end{array}
\tag{11}
$$

where $G(X^{(k)}, \psi(X^{(k)})) = \tilde{G}(X^{(k)} - \psi(X^{(k)}))$ and $\tilde{G}$ is a CNN to be learnt jointly to $\psi$ during the learning stage. Interestingly, this gradient descent embeds the fixed-point algorithm when $\tilde{G}$ is the identity.

Let us denote respectively by $\mathcal{I}_{FP}$ and $\mathcal{I}_G$ the fixed-point and gradient-based NN-based interpolators, which implement $N_I$ iterations of the proposed interpolation updates. Below, $\mathcal{I}_{NN}$ will denote both $\mathcal{I}_{FP}$ and $\mathcal{I}_G$. Whereas $\mathcal{I}_{FP}$ is parameter-free, $\mathcal{I}_G$ involves the parameterization of operator $G$. We typically consider a CNN with ReLu activations with increasing numbers of filter through layers up to the final layer which applies a linear convolutional with a number of filters given by the dimension of the state.

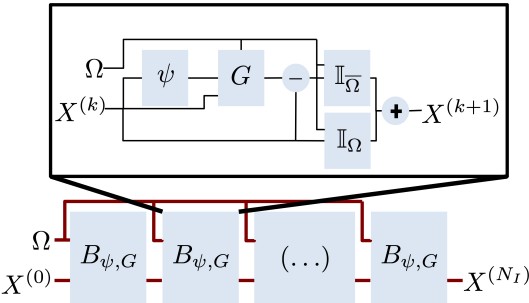

Figure 1: **Sketch of the considered end-to-end architecture**: we depict the considered $N_I$-block architecture which implements a $N_I$-step interpolation algorithm described in Section 3.2. Operator $\psi$ is defined through energy representation (7) and $G$ refers to the NN-based approximation of the gradient-based update for minimization (1). This architecture uses as input a mask $\Omega$ corresponding to the missing-data-free domain and an initial gap-filling $X^{(0)}$ for state $X$. We typically initially fill missing data with zeros for centered and normalized states.

## 3.3 END-TO-END ARCHITECTURE AND IMPLEMENTATION DETAILS

Given the parameterizations for energy $U_\theta$ and the associated NN-based interpolators presented previously, we design an end-to-end learning for energy representation $U_\theta$ and associated interpolator

$\mathcal{I}_{NN}$, which uses as inputs an observed sample $Y^{(i)}$ and the associated missing-data-free domain $\Omega^{(i)}$. Using a normalization preprocessing step, we initially fill missing data with zeros to provide an initial interpolated state to the architecture. We provide a sketch of the architecture in Fig.1.

Regarding implementation details, beyond the design of the architectures, which may be application-dependent for operators $\psi$ and $\tilde{G}$ (see Section 4), we consider an implementation under keras using tensorflow as backend. Regarding the training strategy, we use adam optimizer. We iteratively increase the number of blocks $N_I$ (number of gradient steps) to avoid the training to diverge. Similarly, we decrease the learning rate accross iterations, typically from 1e-3 to 1e-6. In our experiments, we typically consider from 5 to 15 blocks. All the experiments were run under workstations with a single GPU (Nvidia GTX 1080 and GTX 1080 Ti).

## 4 EXPERIMENTS

In this section, we report numerical experiments on different datasets to evaluate and demonstrate the proposed scheme. We consider three different case-studies: an image dataset, namely MNIST; a multivariate time-series through an application to Lorenz-63 dynamics (17) and an image sequence dataset through an application to ocean remote sensing data with real missing data patterns. In all experiments, we refer to the AE-based framework, respectively as FP(d)-ConvAE and G(d)-ConvAE using the fixed-point or gradient-based interpolator where the value of $d$ refers to the number of interpolation steps. Similarly, we refer to the Gibbs-based frameworks respectively as FP(d)-GENN and G(d)-GENN.

### 4.1 MNIST DATASETS

We evaluate the proposed framework on MNIST datasets for which we simulate missing data patterns. The dataset comprises 60000 28x28 grayscale images. For this dataset, we only evaluate the AE-based setting. We consider the following convolutional AE architecture with a 20-dimensional encoding space:

- **Encoder operator** $\phi_E$: Conv2D(20)+ ReLU + AvPooling + Conv2D(40) + ReLU + AveragePooling + Dense(80) + ReLU + Dense(20);

- **Decoder operator** $\phi_E$: Conv2DTranspose(40) + ResNet(2), ResNet: Conv2D(40)+ReLU+Conv2D(20)

We generate random missing data patterns composed of $N_S$ squares of size $W_S x W_S$, the center of the square is randomly sampled uniformly over the image grid. As illustrated in Fig.3, we consider four missing data patterns: $N_S = 20$ and $W_S = 5$, $N_S = 30$ and $W_S = 5$, $N_S = 3$ and $W_S = 9$, $N_S = 6$ and $W_S = 9$. As performance measure, we evaluate an interpolation score (I-score), a global reconstruction score (R-score) for the interpolated images and an auto-encoding (AE-score) score of the trained auto-encoder applied to gap-free data, in terms of explained variance. We also evaluate a classification score (C-score), in terms of mean accurcay, using the 20-dimensional encoding space as feature space for classification with a 3-layer MLP. We report all performance measures for both the test dataset in Tab.1 for MNIST dataset. For benchmarking purposes, we also report the performance of DINEOF framework, which uses a 20-dimensional PCA trained on the gap-free dataset, the auto-encoder architecture trained on gap-free dataset as well as the considered convolutional auto-encoder trained using an initial zero-filling for missing data areas and a training loss computed only of observed data areas. The later can be regarded as a FP(1)-ConvAE architecture using a single block in Fig.1. Overall, these results illustrate that representations trained from gap-free data may not apply when considering significant missing data rates as illustrated by relatively poor performance of PCA-based and AE schemes, when trained from gap-free data. Similarly, training an AE representations using as input a zero-filling strategy lowers the auto-encoding power when applied to gap-free data. Overall, the proposed scheme guarantees a good representation in terms of AE score with an additional gain in terms of interpolation performance, typically between $\approx 15\%$ and 30% depending of the missing data patterns, the gain being greater when considering larger missing data areas.

| New MNIST | Model | I-score | R-score | AE-score | C-score |
|---|---|---|---|---|---|
| $N_S = 30$ | DINEOF | -9.41% (-10.95%) | 21.54% (20.48%) | 64.36% (65.11%) | 96.23% |
| $W = 5$ | ConvAE | 55.98% (55.39%) | 80.98% (80.58%) | **93.42%** **(92.35%)** | **98.12%** |
| | Zero-ConvAE | 61.79% (61.63%) | 82.22% (81.64%) | 87.64% (87.56%) | 97.55% |
| | FP(15)-ConvAE | 74.99% (72.80%) | 88.78% (87.31%) | 91.62% (91.13%) | 97.96% |
| | G(14)-ConvAE | **76.50%** **(75.56%)** | **89.81%** **(88.81%)** | 91.77% (91.21%) | 97.91% |
| $N_S = 30$ | DINEOF | -8.86% (-10.19%) | 13.89% (12.71%) | 64.36% (65.11%) | 96.23% |
| $W = 5$ | ConvAE | 38.32% (38.16%) | 67.42% (67.32%) | **93.42%** **(92.35%)** | **98.12%** |
| | Zero-ConvAE | 53.69% (53.44%) | 74.97% (74.44%) | 85.67% (85.83%) | 97.03% |
| | FP(15)-ConvAE | 69.27% (67.68%) | 83.81% (82.54%) | 90.22% (90.04%) | 97.59% |
| | G(14)-ConvAE | **69.82%** **(68.52%)** | **84.96%** **(83.76%)** | 90.98% (90.66%) | 97.45% |
| $N_S = 3$ | DINEOF | -41.65% (-44.77%) | 33.08% (32.26%) | 64.36% (65.11%) | 96.23% |
| $W = 9$ | ConvAE | -1.21% (-3.08%) | 72.57% (71.96%) | **93.42%** **(92.35%)** | **98.12%** |
| | Zero-ConvAE | 3.55% (1.85%) | 74.04% (72.93%) | 89.21% (89.05%) | 97.76% |
| | FP(15)-ConvAE | **46.91%** **(44.12%)** | 85.13% (83.79%) | 91.87% (91.38%) | 97.90% |
| | G(14)-ConvAE ** | 46.74% (43.76%) | **85.72%** **(83.98%)** | 92.09% (91.39%) | 97.76% |
| $N_S = 6$ | DINEOF | -37.47% (-40.00%) | 16.83% (15.50%) | 64.36% (65.11%) | 96.23% |
| $W = 9$ | ConvAE | -27.02% (-28.28%) | 46.95% (46.44%) | **93.42%** **(92.35%)** | **98.12%** |
| | Zero-ConvAE | -9.94% (-12.03%) | 55.41% (54.09%) | 86.52% (86.73%) | 97.33% |
| | FP(15)-ConvAE | **26.90%** **(22.56%)** | **71.18%** (68.45%) | 91.03% (90.41%) | 97.71% |
| | G(10)-ConvAE | 26.18% (24.73%) | 70.70% **(69.58%)** | 90.30% (90.23%) | 97.86% |

Table 1: **Performance of AE schemes in presence of missing data for Fashion MNIST dataset:** for a given convolutional AE architecture (see main text for details), a PCA and ConvAE models trained on gap-free data with a 15-iteration projection-based interpolation (resp., DINEOF and ConvAE), a zero-filling stratefy with the same ConvAE architecture (Zero-ConvAE) and the fixed-point and gradient-based versions of the proposed scheme. For each experiment, we evaluate four measures: the reconstruction performance for the known image areas (R-score), the interpolation performance for the missing data areas (I-score), the reconstruction performance of the trained AE when applied to gap-free images (AE-score), the classification score of a MLP classifier trained in the trained latent space for training images involving missing data.

## 4.2 MULTIVARIATE TIME SERIES

We present an application to the Lorenz-63 dynamics (17), which involve a 3-dimensional state governed by the following ordinary differential equation:

$$\begin{cases} \frac{dX_{t,1}}{dt} &= \sigma\left(X_{t,2} - X_{t,2}\right) \\ \frac{dX_{t,2}}{dt} &= \rho X_{t,1} - X_{t,2} - X_{t,1}X_{t,3} \\ \frac{dX_{t,3}}{dt} &= X_{t,1}X_{t,2} - \beta X_{t,3} \end{cases} \tag{12}$$

Under parameterization $\sigma = 10$, $\rho = 28$ and $\beta = 8/3$ considered here, Lorenz-63 dynamics are chaotic dynamics, which make then challening in our context. They can be regarded as a reduced-order model of turbulence dynamics. We simulate Lorenz-63 time series of 200 time steps using a Runge-Kutta-4 ODE solver with an integration step of 0.01 from an initial condition in the attractor. For a given experiment, we first subsample the simulated series to a given time step $dt$ and then generate using a uniform random samplong a missing data mask accounting for 75% of the data. Overall, training and test time series are formed by subsequences of 200 time steps. We report experiments with the GE-NN setting. The AE-based framework showed lower performance and is not included here. The considered GE-NN architecture is as follows: a 1D convolution layer with 120 filters with a kernel width of 3, zero-weight-constraints for the center of the convolution kernel and a Relu activation, a 1D convolution layer with 6 filters a kernel width of 1 and a Relu activation, a residual network with 4 residual units using 6 filters with a kernel width of 1 and a linear activation. The last layer is a convolutional layer with 3 filters, a kernel width of 1 and a linear activation.

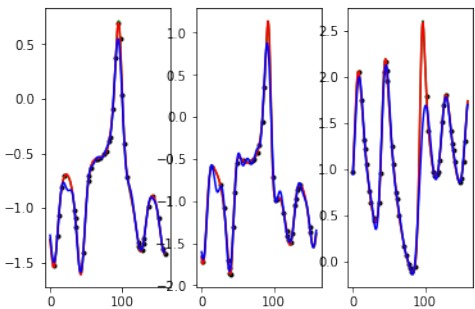

Figure 2: **Example of missing data interpolation for Lorenz-63 dynamics**: from left to right, the time series of each of the three components of Lorenz-63 states for $dt = 0.02$ and a 75% missing data rate. We depict the irregularly-sampled observed data (black dots), the true state (green,-), the interpolated states using DINEOF (blue, -) and the interpolated states using the proposed approach (G-NN-FP-OI) (red, -). Visually, the interpolated sequence using our approach can hardly be distinguished from the true states.

For benchmarking purposes, we report the interpolation performance issued from an ensemble Kalman smoother (EnKS) (10) knowing the true model, regarded as a lower-bound of the interpolation performance. The parameter setting of the EnKS is as follows: 200 members, noise-free dynamical model and spherical observation covariannce to $0.1 \cdot I$. We also compare the proposed approach to DINEOF (2; 21). Here, the learning of the PCA decomposition used in the DINEOF scheme relies on gap-free data. Fig.2 illustrates this comparison for one sequence of 200 time steps with $dt = 0.02$. In this example, one can hardly distinguish the interpolated sequence using the proposed approach (FP(15)-GE-NN). By contrast, DINEOF scheme cannot retrieve some of the largest deviations. We report in Appendix Tab.3 the performance of the different interpolation schemes. The proposed approach clearly outperforms DINEOF by about one order of magnitude for the experiments with a time-step $dt = 0.02$. The interpolation error for observed states (first line in Tab.3) also stresses the improved prior issued from the proposed Gibbs-like energy setting. For chaotic dynamics, global PCA representation seems poorly adapted where local representations as embedded by the considered Gibbs energy setting appear more appealing.

### 4.3 Image sequence dataset with very large missing data rates

The third case-study addresses satellite-derived Sea Surface Temperature (SST) image time series. Due to their sensitivity to the cloud cover, such SST datasets issued from infrared sensors may involve large missing data rates (typically, between 70% and 90%, Fig.**??** for an illustration). For evaluation purposes, we build a groundtruthed dataset from high-resolution numerical simulations, namely NATL60 data (1), using real cloud masks from METOP AVHRR sensor (19). We sample 128x512 images over five consecutive days from June to August at a 0.05°resolution in an open ocean region in the North-East Atlantic from (40.58°N,46.3°W) to (53.04°N, 16.18°W). Overall, we randomly sample 400 128x512x11 image series as training data and 150 as test dataset.

For this case-study, we consider the following four architectures for the AEs and the GE-NNs:

- ConvAE$_1$: the first convolutional auto-encoder involves the following encoder architecture: five consecutive blocks with a Conv2D layer, a ReLu layer and a 2x2 average pooling layer, the first one with 20 filters the following four ones with 40 filters, and a final linear convolutional layer with 20 filters. The output of the encoder is 4x16x20. The decoder involves a Conv2DTranspose layer with ReLu activation for an initial 16x16 upsampling stage a Conv2DTranspose layer with ReLu activation for an additional 2x2 upsampling stage, a Conv2D layer with 16 filters and a last Conv2D layer with 5 filters. All Conv2D layers use 3x3 kernels. Overall, this model involves $\approx$ 400,000 parameters.

- ConvAE$_2$: we consider a more complex auto-encoder with an architecture similar to ConvAE$_1$ where the number of filters is doubled (*e.g.*, The output of the encoder is a 4x16x40 tensor). Overall, this model involves $\approx$ 900,000 parameters.

- GE-NN$_{1,2}$: we consider two GE-NN architectures. They share the same global architecture with an initial 4x4 average pooling, a Conv2D layer with ReLu activation with a zero-weight constraint on the center of the convolution window, a 1x1 Conv2D layer with N filters, a ResNet with a bilinear residual unit, composed of an initial mapping to an initial 32x128x(5*N) space with a Conv2D+ReLu layer, a linear 1x1 Conv2D+ReLu layer with N filters and a final 4x4 Conv2DTranspose layer with a linear activation for an upsampling to the input shape. GE-NN$_1$ and GE-NN$_2$ differ in the convolutional parameters of the first Conv2D layers and in the number of residual units. GE-NN$_1$ involves 5x5 kernels, $N = 20$ and 3 residual units for a total of $\approx$ 30,000 parameters. For GE-NN$_2$, we consider 11x11 kernels, $N = 100$ and 10 residual units for a total of $\approx$ 570,000 parameters.

These different parameterizations were selected so that ConvAE$_1$ and GE-NN$_2$ involve a modeling complexity in the same range. We may point out that the considered GE-NN architecture are not applied to the finest resolution but to downscaled grids by a factor of 4. The application of GE-NNs to the finest resolution showed poor performance. This is regarded as an illustration of the requirement for considering a scale-selection problem when applying a given prior. The upscaling involves the combination of a Conv2DTranspose layer with 11 filters, a Conv2D layer with a ReLu activation with 22 filters and a linear Conv2D layer with 11 filters.

Similarly to MNIST dataset, we report the performance of the different models in terms of interpolation score (I-score), reconstruction score (R-score) and auto-encoding score (AE-score) both for the training and test dataset. We compare the performance of the four models using the fixed-point and gradient-based interpolation. Overall, we can draw conlusions similar to MNIST case-study. Representations trained from gap-free data lead to poor performance and the proposed scheme reaches the best performance (gain over 50% in terms of explained variance for the interpolation and reconstruction score). Here, models trained with a zero-filling strategy show good interpolation and reconstruction performance, but very poor AE score, stressing that cannot apply beyond the considered interpolation task. When comparing GE-NN and AE settings, GE-NNs show slightly better performance with a much lower complexity (e.g., 30,000 parameters for GE-NN$_1$ vs. 400,000 parameters for ConvAE$_1$). Regarding the comparison between the fixed-point and gradient-based interpolation strategies, the later reaches slightly better interpolation and reconstruction score. We may point out the significant gain w.r.t. OI, which is the current operational tool for ocean remote sensing data. We illustrate these results in Appendix (Fig.6), which further stresses the gain w.r.t. OI for the reconstruction of finer-scale structures.

| SST | Model | I-Score | R-score | AE-score |
|-----|-------|---------|---------|----------|
| | OI | 67.59% (57.29%) | 70.97% (61.00%) | - |
| AE models | FP(5)-PCA(20) | 32.52% (39.22%) | 34.94% (30.39%) | 74.17% (56.00%) |
| | FP(5)-PCA(80) | 28.01% (34.83%) | 30.91% (25.28%) | **89.95%** (64.53%) |
| | Zero-ConvAE$_1$ | 89.12% (86.98%) | 89.65% (87.33%) | 67.42% (60.41%) |
| | FP(10)-ConvAE$_1$ | 87.63% (85.24%) | 89.82% (87.28%) | 83.81% (77.20%) |
| | G(8)-ConvAE$_1$ | 89.08% (87.89%) | 89.51% (88.25%) | 84.22% (76.32%) |
| | Zero-ConvAE$_2$ | 86.70% (86.37%) | 87.14% (86.87%) | 67.20% (54.77%) |
| | FP(10)-ConvAE$_2$ | 88.71% (85.02%) | 89.14% (85.49%) | 86.24% **(80.76)** |
| | G(8)-ConvAE$_2$ | 90.47% (88.00%) | 90.98% (88.39%) | 86.33% (78.33%) |
| GE-NN models | Zero-GE-NN$_1$ | 85.46% (79.39%) | 86.71% (80.30%) | -94.84% (-172.68%) |
| | FP(15)-GE-NN$_1$ | 89.22% (87.45%) | 90.07% (88.50%) | 92.61% (90.18%) |
| | G(12)-GE-NN$_1$ | 89.83% **(89.16%)** | 90.56% **(90.00%)** | 92.23% (90.98%) |
| | Zero-GE-NN$_2$ | 86.60% (77.38%) | 87.48% (78.01%) | -141.64% (-235.50%) |
| | FP(15)-GE-NN$_2$ | 90.56% (85.93%) | 91.33% (87.26%) | **93.04%** **(91.17%)** |
| | G(12)-GE-NN$_2$ | **91.10%** (87.98%) | **91.83%** (88.81%) | 92.36% (90.37%) |

Table 2: **Performance on SST dataset:** We evaluate for each model interpolation, reconstruction and auto-encoding scores, resp. I-score, R-score and AE-score, in terms of percentage of explained variance resp. for the interpolation of missing data areas, the reconstruction of the whole image with missing data and the reconstruction of gap-free images. For each model, we evaluate these score for the training data (first row) and the test dataset (second row in brackets). We consider four different auto-encoder models, namely 20 and 80-dimensional PCAs and ConvAE$_{1,2}$ models, and two GE-NN models, GE-NN$_{1,2}$, combined with three interpolation strategies: the classic zero-filling strategy (Zero) and proposed iterative fixed-point (FP) and gradient-based (G) schemes, the figure in brackets denoting the number of iterations. For instance, FP(10)-GE-NN$_1$ refers to GE-NN$_1$ with a 10-step fixed-point interpolation scheme. The PCAs are trained from gap-free data. We also consider an Optimal Interpolation (OI) with a space-time Gaussian covariance with empirically-tuned parameters. We refer the reader to the main text for the detailed parameterization of the considered models.

## 5 CONCLUSION

In this paper, we have addressed the learning of energy-based representations of signals and images from observation datasets involving missing data (with possibly very large missing data rates). Using the proposed architectures, we can jointly learn relevant representations of signals and images while jointly providing the associated interpolation schemes. Our experiments stress that learning representations from gap-free data may lead to representations poorly adapted to the analysis of data with large missing data areas. We have also introduced a Gibbs priors embedded in a neural network architecture. Relying on local characteristics rather than global ones as in AE schemes, these priors involve a much lower complexity. Our experiments support their relevance for addressing inverse problems in signal and image analysis. Future work may further explore multi-scale extensions of the proposed schemes along with couplings between global and local energy representations

and hybrid minimization schemes combining both gradient-based and fixed-point strategies in the considered end-to-end formulation.

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

## A  APPENDIX

### A.1  SUPPLEMENTARY FOR MNIST DATASET

We illustrate below both the considered masking patterns as well as reconstruction examples for the proposed framework applied to MNIST dataset.

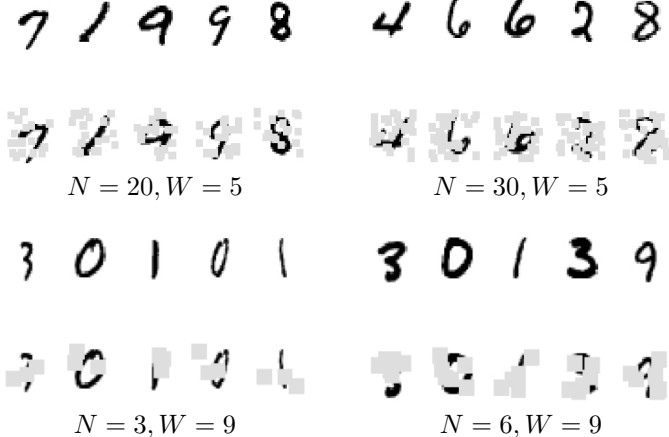

Figure 3: **Illustration of the considered MNIST dataset with the selected missing data patterns:** we randomly remove data from $N$ squared areas of size $W$.

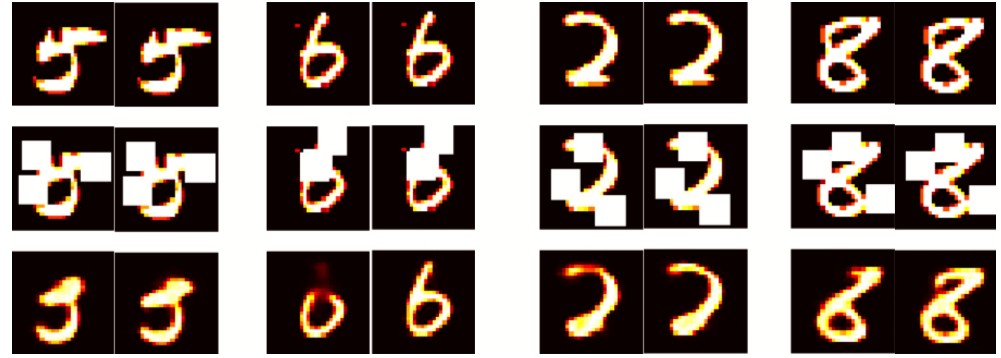

Figure 4: **Illustration of reconstruction results for FP-ConvAE model for MNIST examples:** for each panel, the first column refers to Zero-ConvAE$_1$ results and the second one to Fp(15)-ConvAE$_1$. The first row depicts the reference image, the second row the missing data mask and the third one the interpolated image. The first two panels illustrate interpolation results for training data and last two for test data. We depict grayscale mnist images using false colors to highmight differences.

### A.2  SUPPLEMENTARY FOR LORENZ-63 DYNAMICS

We report below a Table which details the interpolation performance of the proposed GE-NN representation applied to Lorenz-63 time series in comparison with a PCA-based scheme and a lower-bound provided by the interpolation assuming the ODE model is known.

### A.3  SUPPLEMENTARY FOR SST DATASET

We report below reconstruction examples for the application of the proposed GE-NN approach to SST time series with real missing data masks, which involve very large missing data rates (typically above 80%). The consistency between the interpolation results and the reconstruction of the gap-free image from the learnt energy-based representation further stresses the ability of the proposed approach to extract a generic representation from irregularly-sampled data. These reulsts

| Missing data | EnKS | DINEOF | G-NN-FP-OI |
|---|---|---|---|
| 1/4 ($dt = 0.01$) | 9.91e-3 | 5.27E-01 | 5.97e-04 |
| | 2.83e-3 | 1.10e0 | 6.55e-03 |
| 1/4 ($dt = 0.02$) | 6.15e-02 | 4.53e-01 | 2.17e-03 |
| | 1.92E-02 | 4.31e0 | 5.60e-02 |
| 1/4 ($dt = 0.04$) | 5.41e-01 | 5.78e-01 | 7.90e-03 |
| | 4.17e-01 | 1.33e01 | 7.39e-01 |

Table 3: **Interpolation performance for Lorenz-63 dynamics with different missing data rates**: we compare the proposed neural-network aproach (FP(15)-GE-NN) to an ensemble Kalman Smoother (EnKS) assuming the dynamical model (12) is known, and a DINEOF scheme (21). We report interpolation results for a 75% missing data rate with uniform random sampling for three different sampling time steps, $dt = 0.01$, $dt = 0.02$ and $dt = 0.04$. We report the mean square error of the interpolation for the observed data (first row) and masked ones (second row).

also emphasize a much greater ability of the proposed learning-based scheme to reconstruct fine-scale structures, which can hardly be reconstructed by an OI scheme with a Gaussian space-time covariance model. We may recall that the later is the stae-of-the-art approach for the processing of satellite-derived earth observation data (8).

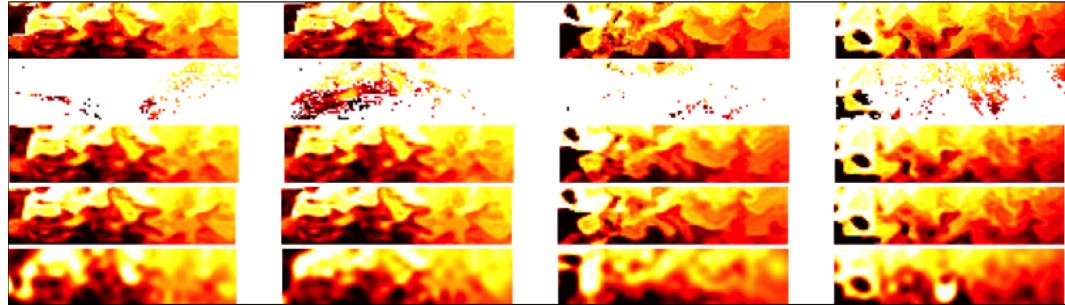

Figure 5: **Interpolation examples for SST data used during training:** first row, reference SST images corresponding to the center of the considered 11-day time window; second row, associated SST observations with missing data, third row, interpolation issued from FP(15)-GE-NN$_2$ model; third row, reconstruction of the gap-free image series issued from FP(15)-GE-NN$_2$ model; interpolation issued from an optimal interpolation scheme using a Gaussian covariance model with empirically tuned parameters.

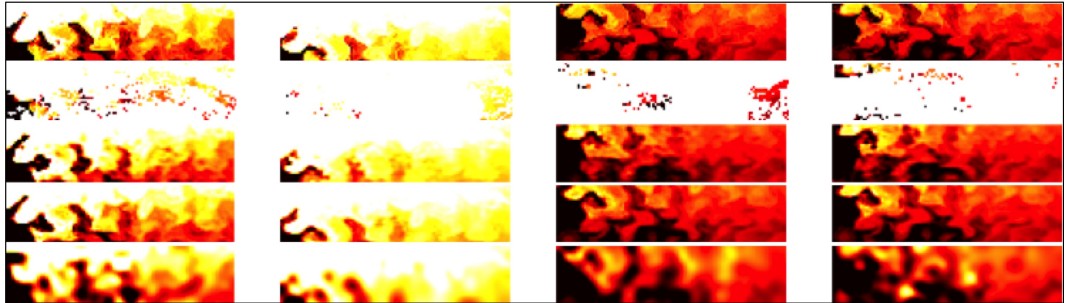

Figure 6: **Interpolation examples for SST data never seen during training:** first row, reference SST images corresponding to the center of the considered 11-day time window; second row, associated SST observations with missing data, third row, interpolation issued from FP(15)-GE-NN$_2$ model; third row, reconstruction of the gap-free image series issued from FP(15)-GE-NN$_2$ model; interpolation issued from an optimal interpolation scheme using a Gaussian covariance model with empirically tuned parameters.

