# OpenReview forum: "End-to-end learning of energy-based representations for irregularly-sampled signals and images"
_ICLR.cc/2020/Conference — Reject_

### Official Review · AnonReviewer1 · 2019-10-24
**Official Blind Review #1**

**Rating:** 3

**Review:**

The paper proposes an end-to-end learning framework for interpolation problems, motivated by problems such as irregularly-sampled images or time-series.

It was not clear after reading the paper where the key novelty of the proposal lies. The energy formulations for Uθ, namely an autoencoder and Gibbs model, are not to my knowledge new in this context. I gather the novelty must then be in Section 3.2, which uses a neural-network based interpolation scheme. This builds on an existing update scheme (Equation 8), but replaces the full operator ψ with an iterative update. The key idea is then to replace the full gradient for the energy function with a learned function, based on a CNN. While this is not my area of expertise, I am not sure as to technical significance or novelty of such a proposal.

The paper emphasises their view of an interpolation operator I, which is a little confusing in an autoencoder context. Typically, we pick \hat{X} to be the solution attaining minimal squared error compared to the reference Y. It was not clear why one needs a further invocation of a minimisation between Y^i and I(Uθ, Y^i, Ω^i).

The writing could be improved, as there are a number of grammatical issues, as well as missing section or equation #'s. In terms of presentation, the Introduction delves far too soon into a detailed discussion of related work in the area. I would suggest instead to provide a crisp high-level overview of the limitations of existing work, and how they are overcome in the present paper.

**Experience Assessment:**

I do not know much about this area.

**Review Assessment: Checking Correctness Of Derivations And Theory:**

I assessed the sensibility of the derivations and theory.

**Review Assessment: Checking Correctness Of Experiments:**

I assessed the sensibility of the experiments.

**Review Assessment: Thoroughness In Paper Reading:**

I read the paper at least twice and used my best judgement in assessing the paper.

---

### Official Review · AnonReviewer3 · 2019-10-29
**Official Blind Review #3**

**Rating:** 1

**Review:**

The paper proposes a framework to learn representations of signals when they the signals are irregularly sampled.  They propose to do this by using some modified iteration steps from DINEOF algorithm. In addition to this, they propose a new energy function which is inspired by the idea of Gibbs distribution. The parametrize the energy function by some convolutional filters with a constraint. As claimed by the paper in the introduction, they only use the under sampled signals to learn as opposed to using the fully sampled ground truth signals in the previous deep learning basic approaches.

I choose to reject this paper. 1) The paper doesn't justify what these representations are or where these representations will be used 2) There is no theoretical or extensive empirical motivation that the representation that is coming out of this network is actually useful for downstream tasks of interest. 3) Ill-defined evaluation metrics 4) Missing comparisons with basic models - why not compare with in-painting models for images and time series in experiments? 5) The paper is hard to read, not well organized and missing a lot of details.

Main Argument:
It is not clear to me the motivation for learning these representations or how the usefulness of this representation will be evaluated. The only place where I see a use for representation other than reconstruction of the signal is in Table 1. They use a 3 layered MLP on top of the representation from auto encoder. Why is this interesting? If you're trying to claim that the representation is powerful enough, shouldn't you be able to do the classification with just a linear classifier? Can you justify the usefulness of representation for some downstream tasks in your other experiments?

In any case, I don't understand why the authors don't compare with a simple auto-encoder. The authors assumes that they know where the signal is sampled and where is is not. Why not train an auto-encoder with MSE loss, but calculating MSE over only those pixels which the authors know are sampled. Comparison with such a model will be helpful in understanding if the framework is adding anything of value.

Authors useful I-Score, R-Score, AE-Score and C-Score to compare across models. These metrics are not defined in the main text (or any reference) but there is a loose (english) definition in the caption of Table 1. For example, R-Score, is the "reconstruction performance of known image areas". Why are these numbers in percentages? Shouldn't you be using MSE, PSNR or SSIM? Why are some of scores negative?

If you are comparing how well you're interpolating for natural images or time series, you should compare with standard interpolation techniques in image processing and time series analysis.

The paper have a lot of missing details, including simple things like not linking to a Figure or Section the authors are referring to. Some examples are:
1. Authors motivate using Gibbs energy by saying that in auto encoders you have to project to a lower dimensional representation and gibbs energy based solution has no such dimensionality reduction constraint. There are no such constraint in AE as well - you can vey well have your hidden representation to be over complete - so this is not a good motivation for Gibbs energy
2. Main section of 4.1 says that they use MNIST but the caption of Table 1 says they use Fashion MNIST
3. The naming convention of the model is hard to follow and the authors notation itself is inconsistent.

Some other problems:
1. In Abstract, they say 2. images. Why 2?
2. Introduction, the citation for Gaussian priors is empty
3. Last word before beginning of section to, broken link
4. In the second para of 3.2 did you mean to say solve for (2) instead of (3)?
5.  What is "XXX" matrix in second para of 3.2?




**Experience Assessment:**

I do not know much about this area.

**Review Assessment: Checking Correctness Of Derivations And Theory:**

N/A

**Review Assessment: Checking Correctness Of Experiments:**

I assessed the sensibility of the experiments.

**Review Assessment: Thoroughness In Paper Reading:**

I read the paper thoroughly.

---

### Official Review · AnonReviewer4 · 2019-10-31
**Official Blind Review #4**

**Rating:** 1

**Review:**

This paper discusses various approaches to predict missing values in the input (filling/inpainting task).
They define an energy function equal to the squared L2 distance between the input and its reconstruction by various kinds of neural nets. They show slightly better performance compared to a PCA-based baseline.

Positive things about this work
- the topic is interesting
- the last application is interesting (water temperature prediction)

Negative things about this work
- this work is very poorly  written and lacks sufficient clarity. This work needs a major rewrite. I do not even know where to start, but to give an example:
1st sentence  of the abstract reads:  “For numerous domains, including for instance earth observation, medical imaging, astrophysics,..., available image and signal datasets often irregular spacetime sampling patterns and large missing data rates.”, a sentence that misses the verb.
2nd sentence of the abstract reads: "These sampling properties is” which is not grammatically correct
And so on so forth. Speaking of which, the Authors make excessive use of "....".
- because of the lack of clarity throughout the paper, I have had hard time figuring out what exactly the authors do. From the limited understanding I got after reading this draft twice, I think they consider a few different variants of auto-encoder and a "log-prior"/energy function equal to the squared L2 distance between input and its reconstruction. However, this formulation has barely any novelty. For instance, see old work like:
S. Roth and M. J. Black, “Fields of experts: A framework for learning image priors,” in Proc. of the IEEE Conference on Computer Vision and Pattern Recognition (CVPR), vol. 2, San Diego, California, June 2005, pp. 860–867
where they used a different log-prior, but essentially the same optimization process. If the novelty is the use of auto-encoders, then the comparison should be against methods that do not auto-encode (like the above or more modern versions of it).
- several choices made by the Authors seem not well motivated. For instance, it's written that computing gradients is too expensive and therefore these are replaced by the G network. However, doesn't the G network also need gradients to be updated?
- the terminology is not standard and confusing. Hidden state usually refers to the encoder output, not to the decoder output.
- the Authors never introduce the metrics they use, reconstruction and interpolation scores.
- in general, the motivation is unclear. Why is it a problem if the data does not come from a grid? In vision, inpainting on unconstrained masks has been standard for decades.
More recently, transformer architectures and GNNs seem quite good at representing sets and graph structured data.
So considering this context, the current motivation provided by the authors needs some refinement. In fact, the authors could consider building on top of these other more modern approaches.

**Experience Assessment:**

I have published one or two papers in this area.

**Review Assessment: Checking Correctness Of Derivations And Theory:**

I assessed the sensibility of the derivations and theory.

**Review Assessment: Checking Correctness Of Experiments:**

I assessed the sensibility of the experiments.

**Review Assessment: Thoroughness In Paper Reading:**

I read the paper at least twice and used my best judgement in assessing the paper.

---

### Decision · Program_Chairs · 2019-12-19

**Decision:**

Reject

**Comment:**

This work looks at ways to fill in incomplete data, through two different energy terms.
Reviewers find the work interesting, however it is very poorly written and nowhere near ready for publication. This comes on top of poorly stated motivation and insufficient comparison to prior work.
Authors have chosen not to answer the reviewers' comments.
We recommend rejection.